# Molecular Conditional Generation and Property Analysis of Non-Fullerene Acceptors with Deep Learning

**DOI:** 10.3390/ijms22169099

**Published:** 2021-08-23

**Authors:** Shi-Ping Peng, Xin-Yu Yang, Yi Zhao

**Affiliations:** State Key Laboratory for Physical Chemistry of Solid Surfaces, Fujian Provincial Key Lab of Theoretical and Computational Chemistry, College of Chemistry and Chemical Engineering, Xiamen University, Xiamen 361005, China; psphi@stu.xmu.edu.cn (S.-P.P.); 20520181152664@stu.xmu.edu.cn (X.-Y.Y.)

**Keywords:** non-fullerene acceptors, convolutional neural networks, molecular generation model, frontier molecular orbital energies, chemical space

## Abstract

The proposition of non-fullerene acceptors (NFAs) in organic solar cells has made great progress in the raise of power conversion efficiency, and it also broadens the ways for searching and designing new acceptor molecules. In this work, the design of novel NFAs with required properties is performed with the conditional generative model constructed from a convolutional neural network (CNN). The temporal CNN is firstly trained to be a good string-based molecular conditional generative model to directly generate the desired molecules. The reliability of generated molecular properties is then demonstrated by a graph-based prediction model and evaluated with quantum chemical calculations. Specifically, the global attention mechanism is incorporated in the prediction model to pool the extracted information of molecular structures and provide interpretability. By combining the generative and prediction models, thousands of NFAs with required frontier molecular orbital energies are generated. The generated new molecules essentially explore the chemical space and enrich the database of transformation rules for molecular design. The conditional generation model can also be trained to generate the molecules from molecular fragments, and the contribution of molecular fragments to the properties is subsequently predicted by the prediction model.

## 1. Introduction

Organic solar cells (OSCs) [1], one of the most promising directions for exploiting solar energy, consist of a blend of electron donor and acceptor materials. Traditionally, fullerene molecules are taken as acceptors because of their high electron affinity and mobility. For further improvement of power conversion efficiency (PCE), non-fullerene acceptors (NFAs) have recently been proposed as alternative acceptors [2]. Compared with fullerenes, NFAs have the advantages of adjustable energy level, appropriate optical absorption, good chemical stability, excellent morphology stability, and low synthesis cost, which could make them the most potential components in organic photovoltaic materials [3,4]. The NFAs have made remarkable progress in the performance of OSCs. By 2020, the PCE of Y6-based OSCs have already achieved over 18% [5]. Furthermore, in some specific applications, such as semi-transparent organic photovoltaic materials for buildings and agriculture and organic photodetectors, NFAs show promising application prospects. One of the kernel characteristics of NFAs is adjustable frontier orbital energies, especially for the highest occupied molecular orbital (HOMO) energy and the lowest unoccupied molecular orbital (LUMO) energy, which play significant roles in modulating the energy level alignments in the donor–acceptor pairs interface in order to enhance the interfacial charge separation efficiency and decrease the voltage losses [6]. However, it is still challenging to find and design the NFAs with desired orbital energies.

Machine learning is a branch of artificial intelligence. Machine learning models are trained for mining hidden insights and internal relationships from data and making classifications, predictions and generations. The past decade has seen increasingly rapid advances in the field of molecular design with machine learning [7]. As one of the most powerful tools for generating new data and approximating intractable functions, deep learning, a sub-field of machine learning, is widely applied in automatic molecular generation [8,9,10,11,12] and property prediction [13,14,15,16,17,18,19]. The sequence-based and graph-based generative models are the most widely used for molecular generation [20,21,22,23,24,25,26], in which the molecules are represented by simplified molecular input line entry specification (SMILES) [27] strings and molecular graphs. To constrain the models for generating molecules with desired properties, the goal-directed molecular generation becomes an increasingly important area in molecular design. Many methods, such as variational autoencoder (VAE) [28,29,30], generative adversarial network (GAN) [31], reinforcement learning (RL) [32,33], transfer learning (TL) [34,35,36], and recurrent neural network (RNN) [37] have been developed for conditional molecular generation. The VAE and GAN can build the relationship between molecular properties and the latent distribution of molecules. RL often combines with GAN to constrain the model for generating the targeted molecules. TL can fine-tune the trained models with smaller sets for generating focused molecules. RNN aggregates selected molecular descriptors along with bioactivity label as initial states for drug design. Most of the methods have RNN-based architectures in sequence models. Although many improvements for RNNs have been proposed [38,39,40], the constraints in parallelization and long-range context processing still remain. The architectures, such as temporal convolutional [41] or attention mechanism [42] are researched for replacing the RNNs.

The past decade has seen increasingly rapid advances in the field of molecular design [7] as well as the NFA design. The Harvard Clean Energy Project in conjunction with the IBM World Community Grid has screened over 51,000 NFAs to identify new classes and fragment ligation patterns [43]. Lee has developed the machine-learning model for predicting the efficiency of non-fullerene OSCs from frontier molecular orbital energy levels of organic materials [44]. Min et al. have introduced 565 donor/acceptor combinations in non-fullerene OSCs as a training dataset to evaluate the practicalities and the application potential of five common ML algorithms [45]. Troisi et al. have considered a database of known organic semiconductors and identified those compounds with computed electronic properties as promising NFAs [46]. Wang et al. have used machine learning to design NFAs for P3HT-based OSCs [47]. Compared with those works, our present work focuses on designing NFAs for given properties and analyzing the structure–property relationships of molecular fragments.

In our previous work [48], we used convolutional neural networks (CNNs) to construct generative and prediction models for the design and analysis of NFAs. It is demonstrated that the causal dilated CNN model is powerful for molecular generation and shows performance as good as other models at the benchmarks of Molecular Sets [26]. The purpose of the present work is to extend our previous model to a conditional generative model for generating the molecules from property distributions directly. Compared with the proposed molecular conditional generative models, the present model uses multi-property as the initial states for material design and has simpler architecture and good performance. Moreover, we further investigate the ability of the generative model for exploring the chemical space. In the model, the molecules are still represented by SMILES strings. Benefiting from the rapid progress in natural language processing, the SMILES-based models show better performance in conditional generation than graph-based models [49].

To judge whether the generated new molecules indeed have required properties, one can verify the results and analyze the structure–property relationship using the SMILES-based prediction model [48]. Many investigations show that the prediction models based on graph neural networks (GNNs) should be a natural and intuitive choice, and they also have excellent performance in the property prediction [13,14,15,16,17,18,19]. These GNNs include message passing neural network [13], deep tensor neural network [14], and so on. In the graph-based models, atoms and bonds are taken as nodes and edges respectively, the interactions of atoms are updated around nodes and edges. In the present work, we alternatively choose GNN-based prediction models to analyze the frontier orbital energies of NFAs. The atomic number and the message about whether bonding or not are provided as inputs, graph attention networks (GAT) [50] are adopted as the main architecture, and the global attention mechanism as the readout phase.

With the help of the presented models, thousands of NFAs with targeted properties will be easily generated and screened out. It can be proved that the generated new molecules can expand the chemical space and provide more possible structural transformations. In addition, the conditional generation of molecules can also start with molecular fragments. The prediction model can accurately get the properties of generated molecules as well as the contributions of molecular fragments to those properties.

## 2. Methods and Materials

### 2.1. Molecular Conditional Generative Model

Figure 1a displays the molecular conditional generative model, which is based on our previous work [48]. The inputs of CNN are described by
(1)$$z_{l}^{0}=\mathrm{Embed}\left(x_{l}\right)∥f_{\Phi }\left(y_{1},\ldots ,y_{n}\right).$$

Here, y1,…,yn, the targeted properties, are transformed to vectors with a hidden size by linear transformation fΦ. The xl is the index of the *l*-th character in the SMILES string, and it is mapped to the vector of real numbers by an embedding layer. The ‖ represents the concatenation of two vectors. These operations are performed on all inputs x0…,xL−1 before entering the convolutional layers. It should be noted that x0 is the index of the addition starting character “&” and xL is the index of the addition ending character “\n” in the SMILES string. For the generation of a string, the hidden state zlt of each character in the *t*-th layer should only relate to the states z0t−1,…,zlt−1 of previous characters. Thus, the causal convolutions are used in the model instead of normal convolutions. The diagram above in Figure 1b is the normal convolution, where two convolutional kernels are applied over one-dimensional multi-channel inputs. The input arrays in the window and each kernel are multiplied and summed by the element to get the element at the corresponding location in the output array. The zero-padding is added to two sides of the input arrays to keep the output arrays having the same length as the input arrays. The zlt is related to zl−1t−1, zlt−1 and zl+1t−1. However, in the causal convolution shown in the diagram below in the Figure 1b, the zero-padding with size 2 is added to the left side of the input arrays, thus the output z0t is only related to zl−2t−1, zl−1t−1 and zlt−1. To incorporate longer messages of preceding characters and reduce the information redundancy, the dilated convolution is thus adopted to increase the spacing between the points of each kernel. In our model, the convolutional kernel size of CNN is taken as 3, the stride of the kernel is 1, the dilation factor is 2t−1 for the *t*-th layer, and the padding size is (k−1)d. After the causal convolution layer, the gated linear unit and residual networks are adopted for multichannel information transmission, and can be written as
(2)Zt+1=GLU(Conv1d(Zt))+Zt,
where Zt represents the inputs of the *t*-th layer z0t,…,zL−1t, the GLU is the gated linear unit, which splits the matrix in half along a given dimension to form a and b matrix, then computes a⊗σ(b), where σ is the sigmoid function and ⊗ is the element-wise product between matrices. The operations are repeated in T layers as shown in the dashed box of Figure 1a. Then, the output of the last convolutional layer, zlT, is decoded to the vector zl with the size of the dictionary (the classes of all characters in the dataset of SMILES strings) by a linear transformation. The zl contains raw and unnormalized scores for each class of character. The criterion of each output is given by the cross entropy loss function
(3)lossl+1=−lnexp(zl,xl+1)∑jexp(zl,j),
where the zl,j is the *j*-th value of vector zl. The loss function aims to maximize the value zl,xl+1 in the zl, where xl+1 is the index of the next character. The losses of all outputs are averaged across a string.

The CNN allows parallelization over every character in a SMILES string with high efficiency during training. However, each molecular SMILES string should be generated character by character, as shown in Figure 1c. The first character is generated by taking “&” as input and the generative process is terminated as the index of “\n” is met. Between them, the character index *j* is sampled from the multinomial probability distribution, and the corresponding probability is given by
(4)$$p_{j}=\frac{\exp(z_{l,j})}{\sum_{j}\exp\left(z_{l,j}\right)}.$$

The posterior probability of the generated molecular SMILES string can be described as the product of the conditional probabilities
(5)$$P_{\theta }\left(x_{1},\ldots ,x_{L}|y_{1},\ldots ,y_{n}\right)=\prod_{l=0}^{L-1}P_{\theta }\left(x_{l+1}|x_{0},\ldots ,x_{l},y_{1},\ldots ,y_{n}\right).$$

### 2.2. A Graph-Based Property Prediction Model

Figure 2 exhibits the architecture of a graph-based model constructed for property prediction in the present work. The input is the atomic number $c_{i}$ of each atom (node), and whether bonding or not between atoms as implicit conditions is also incorporated in the model. Then the node feature $c_{i}$ is transformed to vectors $h_{i}^{0}$ as an initial state (hydrogen atoms are neglected in this work) by an embedding layer before entering the GAT (or other graph neural networks, the detailed architecture of GAT is provided in Appendix B). The global attention network is performed on the output $h_{i}^{T}$ of GAT. The attention coefficient $a_{i}$ of each node for output can be calculated across all nodes by the softmax function
(6)$$a_{i}=\frac{\exp(f_{\Psi }\left(h_{i}^{T}\right))}{\sum_{k\in N}\exp\left(f_{\Psi }\left(h_{k}^{T}\right)\right)},$$
where $f_{\Psi }$ is the linear transformation neural network. Then the output *Y* is achieved by aggregating messages of all nodes *N*
(7)$$Y=\sum_{i\in N}a_{i}⊙f_{\Theta }\left(h_{i}^{T}\right),$$
where the hidden state $h_{i}^{T}$ of each node is transformed to the size of output by linear transformation $f_{\Theta }$ and the ⊙ is element-wise multiplication. Finally, the weight of each atom to the corresponding property can be calculated by
(8)$$w_{i}=\frac{a_{i}⊙f_{\Theta }\left(h_{i}^{\prime }\right)}{Y}.$$

### 2.3. Dataset and Technique Details

The library of NFAs was introduced by Lopez et al. [43]. This library has more than 51,000 molecules, built by the combinations of 12 core fragments, 43 spacer fragments, and 47 terminal fragments. The library provides the theoretical calculated and experimental calibrated data of molecular features, such as HOMO and LUMO energies, and the PCE of solar cell with poly[N-9’-heptadecanyl-2,7-carbazole-alt-5,5-(4’,7’-di-2-thienyl-2’,1’,3’-benzothia-diazole)] as the donor. Figure 3 shows the HOMO and LUMO distributions of the most molecules in the library. In this work, the molecular conditional generative model aims to generate desired molecules with given HOMO and LUMO energies, and the prediction model is a multi-task learning model trained for the prediction of both HOMO and LUMO energies. The filtered dataset used for the generative model contains 43,497 acceptors with the calculated HOMO energies from −7.0 eV to −5.2 eV and LUMO energies from −4.0 eV to −2.2 eV, as shown in the black frame of Figure 3. Most molecules in the dataset for the generative model have band gaps from 2 eV to 4 eV. The dataset is randomly divided into training set, validation set, and test set with sizes of 33,497, 5000, and 5000, respectively. The prediction model is trained with 50,656 acceptors whose HOMO energies locate from −7.6 eV to −4.6 eV and LUMO energies locate from −4.6 eV to −1.6 eV. Here, 40,656 molecules are used for the training set, 5000 for the validation set, and 5000 for the test set.

The deep learning models are built with PyTorch(v1.7) [51] and DGL-LifeSci [52]. DGL-LifeSci is a python toolkit based on RDKit [53], PyTorch, and Deep Graph Library (DGL) [54]. The RDKit is an open-source tool used for related operations of cheminformatics and DGL is an open-source domain package specifically designed for researchers and application developers of GNNs. In the dataset for the generative model, the SMILES strings include 28 unique characters (exclude “&” and “\n”). In our model testing, it is found that the most suitable size of embedding layer for the best performance is 32. For the prediction model, the SMILES strings are transferred to the graph representation with the help of DGL-LifeSci. The models in this work are both trained on NVIDIA 1080Ti with a batch size of 32. Adam is taken as an optimizer with the initializing learning rate of 0.001. The model parameters with the best performance in validation set are saved during 300 epochs of training. With the trained generative model, the mean cross entropy for the test set is 0.0880. With the trained prediction model, the mean-absolute-errors/root-mean-square-errors of HOMO and LUMO energy prediction for the test set are 0.0566 eV/0.0814 eV and 0.0581 eV/0.0889 eV, respectively. More details and codes are available at https://github.com/PSPhi/CGEN-GPRE (last accessed date 20 August 2021).

The new molecules generated may have new structures for designing molecules. Thus, we further use the mmpdb [55], an open-source platform, for matched molecular pair (MMP) analysis. An MMP is formed by two molecules that differ by a defined structural transformation [56], and the mmpdb can compile all possible structural transformation rules for the dataset to build the rule database and guide molecular transformation.

The quantum chemistry calculations of HOMO and LUMO energies are performed by Gaussian 16 [57] at a level of DFT/B3LYP/Def2SVP.

## 3. Results

### 3.1. Conditional Molecular Generation and Evaluation

Before applying the present generative model to NFAs generation, we first evaluate it with GuacaMol, an evaluation framework for de novo molecular design [49]. The benchmarks and performance of the model are provided in Appendix C. For the CNN-based conditional generative model trained by the dataset of NFAs, we first test its ability for generating the molecules with the properties located in the range of the training set. The 4000 targeted HOMO and LUMO energies are randomly chosen from a uniform distribution to generate the molecular SMILES strings. Then the generated SMILES strings are transformed to the molecular graph representation for the prediction of the HOMO and LUMO energies from the prediction model. Figure 4a,b displays the predicted HOMO (LUMO) energies versus targeted HOMO (LUMO) energies, in which there are 3610 reasonable molecules in the 4000 generated SMILES strings. It is seen that the targeted and predicted HOMO (LUMO) energies of the generated molecules have a relative coefficient $R^{2}$ of 0.84 (0.82). In the figures we also color the nearest neighbor molecular similarity for generated molecules. The result shows that most of the generated molecules are similar to the molecules in the dataset. To find the molecules whose predicted HOMO/LUMO energies are closest to the targeted energies, the generation processes are run 30 times (with 120,000 SMILES strings generated). Figure 4c,d show the results of 4000 generated molecules with the smallest mean absolute error (MAE) of HOMO and LUMO energies. The value of $R^{2}$ reaches to 0.99, manifesting that the molecules screened out almost meet the demand. However, the similarity shown in Figure 4c,d indicates that most of the matched molecules come from the dataset. To explore the chemical space, the generated new molecules are selected in 30 runs and 3995 best matched new molecules are screened out for 4000 desired orbital energies, and the corresponding energies are shown in Figure 4e,f. The $R^{2}$ of 0.96 indicates that the conditional molecular generation model also has an excellent performance in generating required new molecules.

Encouraged by these results, we save more than 24,000 new molecules generated here with predicted HOMO/LUMO energies in an extending library and provide it in the Appendix A. This new library may be further used for exploring the chemical space. In the following section, we will demonstrate its potential applications.

### 3.2. Chemical Space Exploring

We first demonstrate the ability of the conditional generative model for exploring chemical space. In the training set, the HOMO and LUMO energies are from −7.0 eV to −5.2 eV and from −4.0 eV to −2.2 eV, respectively. However, we generate molecules with the energy ranges of HOMO and LUMO larger than those in the training set, i.e., the HOMO and LUMO energies are in the range from −7.6 eV to −4.6 eV and from −4.6 eV to −1.6 eV, respectively. For a given HOMO/LUMO energy, we generate 30 SMILES strings with the conditional generative model. Figure 5a displays the MAE of predicted HOMO/LUMO and targeted HOMO/LUMO of the reasonable molecules transformed from 30 generated SMILES strings. As expected, the orbital energies of most generated molecules are close to the targeted values, but, those outside of the training set have obvious deviations, as shown on the outside of the black frame. Furthermore, in the generated molecules from 30 runs, we screen out the best-matched molecules (the predicted HOMO and LUMO energies closest to the targeted properties) and show their corresponding MAEs in Figure 5. The results indicate that the model indeed generates the molecules with required energies if these energies are not too far from the range of the training set. Specially, the generated molecules with bandgaps from 1 eV to 5 eV have low MAEs. To further reveal the ability for generating new molecules, we filter out the best matched new molecules from the molecules shown in Figure 5a, and the corresponding MAEs are shown in Figure 5c. Obviously, the MAEs of some new molecules are larger than those shown in Figure 5b.

Although new molecules may have other energies than the desired ones, they are helpful to explore the chemical space. To do so, we use “mmpdb” to build the structural transformation rule database from the original NFAs library, also the database together with our new NFAs library. Then, we apply the individual structural transformation rules from the two databases to the new molecules in Figure 5c. After that, many potential new molecules are constructed and the orbital energies of these new molecules can be predicted by the prediction model. To see the derivation of these energies to the targeted values, we show the MAE of the molecules transformed with the original rule database in Figure 5d (the MAE from the original database together with ours have a similar behavior, which is not shown). Although the energies of transformed molecules shown in Figure 5d are distributed in a broad range, the energy errors of the best matched new molecules are smaller than the initial molecules in Figure 5c, as shown in Figure 5e, where the MAEs become comparable with those shown in Figure 5b. It is noticed that there are no transformation rules for several molecules shown by white spaces in Figure 5d,e. However, the absent transformation rules can be built up from the combination of original and our NFA libraries. For instance, the molecule at −4.6 eV (HOMO) and −3.6 eV (LUMO) appears, as shown in Figure 5f, where the best matched new molecules (marked by fork) are screened out from transformed molecules with the use of the new transformation rules database. The lower MAEs of molecules in Figure 5f show that more molecules transformed with the new database can also provide more desired molecules.

### 3.3. Fragment-Based Molecular Conditional Generation

The generative model can also be trained with datasets of randomized SMILES strings. With the trained model, the generation of molecules can start with fragments. Figure 6 shows the five possible molecules generated for the targeted HOMO energy of −6.5 eV and LUMO energy of −3.3 eV by the same fragment. The SMILES string of the fragment is “N#Cc1ccc2c(c1)ccs2” in its canonical form. As shown in Table 1, the strings colored in orange are the SMILES specify the last atoms with the help of Open Babel [58]. The five new molecules are generated by the trained model according to the previous characters and the targeted properties. The bonding at different geometric positions can overcome the limitations of the inherent linking relationship between fragments, and enlarge the chemical space. However, the properties of some generated molecules may differ from the targeted properties. The generated molecules can be used as the scaffolds for further decoration. Here, we choose the first molecule in Figure 6 as an example. As shown in Figure 7a, there are also five possible atoms for further decoration. Here, we specify the atom pointed by arrow 2’ as an example and the SMILES string for subsequent generation is underlined. The molecule generated for the targeted HOMO energy of −6.5 eV and LUMO energy of −3.3 eV is displayed in Figure 7b. The results listed in Table 2 exhibit that the frontier orbital energies of molecule (b) are closer to the target energies.

It is significant to understand the molecular structure–property relationship in molecular design. In the present case, this relationship corresponds to the contributions of molecular fragments to the orbital energies, and it can be revealed by the global attention mechanism adopted in the predicted model. In Appendix A, we list the results for some generated new molecules. Figure 7 displays the distributions of HOMO and LUMO obtained by Gaussian 16 calculations as the references for the verification of the predicted results. Table 2 lists the energies of HOMO and LUMO and the weighted sum of each fragment (the column header corresponds to the color of molecular fragment in the left of Figure 7) to the HOMO and LUMO energies from the predicted model. It is seen that the predicted and calculated HOMO and LUMO energies are quite close to each other. The most important fragment from the prediction model and the one from the calculation are also consistent, as shown in the table. In molecules (a) and (b), the fragment colored by orange dominantly determines HOMO energy levels, whereas the LUMO energy level-determined fragment in molecules is colored yellow. It is noticed that the prediction model can correctly predict the weights of fragments to the molecular orbital energies. The red fragment shown in Figure 7b has various weights to HOMO and LUMO. Compared with the variation of HOMO energy, the red fragment in molecule (b) introduces a larger variation of LUMO energy with a larger weight.

## 4. Conclusions

We have extended our previous generative model based on convolutional neural networks to incorporate the properties as conditions for generating the SMILES strings of novel molecules. The proposed conditional generative model is trained to directly generate new non-fullerene molecules with desired HOMO and LUMO energies as initial inputs. The orbital energies of generated molecules are further verified by the graph-based prediction model with the attention mechanism introduced. Together with proposed generative and prediction models trained with the library of NFAs, we have built up a new library with more than twenty thousand new non-fullerene molecules. It is found that the conditional generative model can generate new molecules with an ability to explore chemical space, and the new molecules can be used to generate the database of molecular structural transformation rules to further construct new molecules for screening. Molecular fragments can also be taken as initial conditions for generating needed molecules with a generative model trained by a dataset of randomized SMILES strings. In terms of property prediction, the prediction model predicts the HOMO and LUMO energies correctly. Furthermore, the fragment weights to these orbital energies are successfully predicted by the global attention mechanism in molecular generation processes. It should be mentioned that the HOMO and LUMO energies of new molecules in the present work are predicted by the prediction model, and the accuracy thus heavily depends on the transferability of the prediction model. To improve the accuracy, one should calibrate the orbital energies of generated new molecules with quantum chemistry calculations or experimental measurements. We believe that the architecture of CNN can be seen as an alternative to RNN in various conditional generative models and the information from interpretable models have the potential to further improve the existing models. Besides the HOMO and LUMO energies, solubility, steric hindrance, and electronic properties of NFAs, such as oscillator strength of the excited state and reorganization energy for electron transfer, are also significant for the design of NFAs. These factors can be used as additional properties for further screening if the corresponding datasets are available.

## Figures and Tables

**Figure 1 ijms-22-09099-f001:**
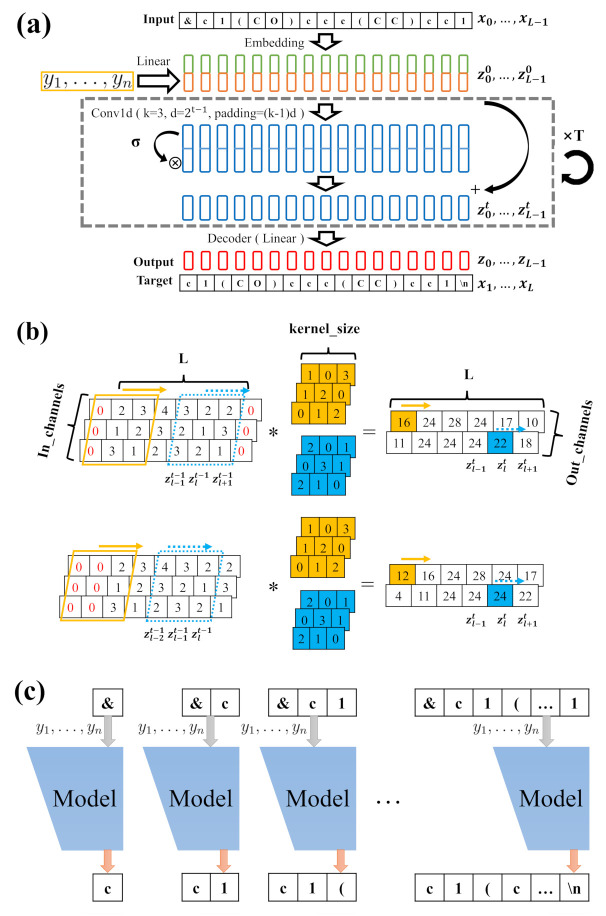
(**a**) The architecture of the generative model. (**b**) The normal one-dimensional multi-channel convolution and causal convolution. (**c**) The conditional generative process of one SMILES string.

**Figure 2 ijms-22-09099-f002:**
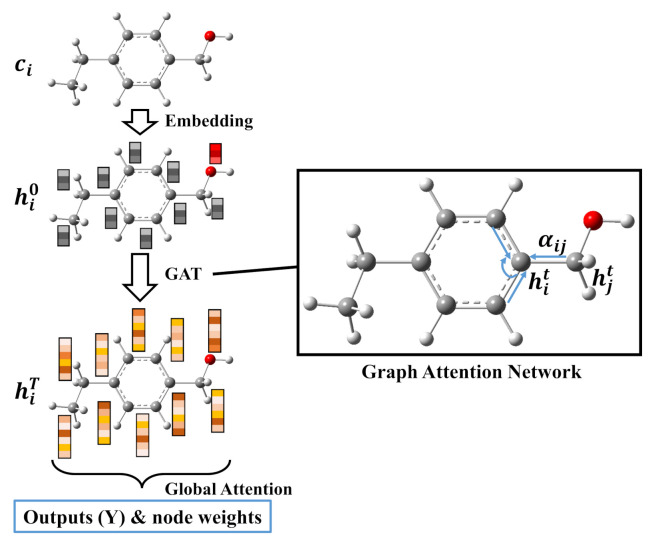
The architecture of the attention-based graph neural network.

**Figure 3 ijms-22-09099-f003:**
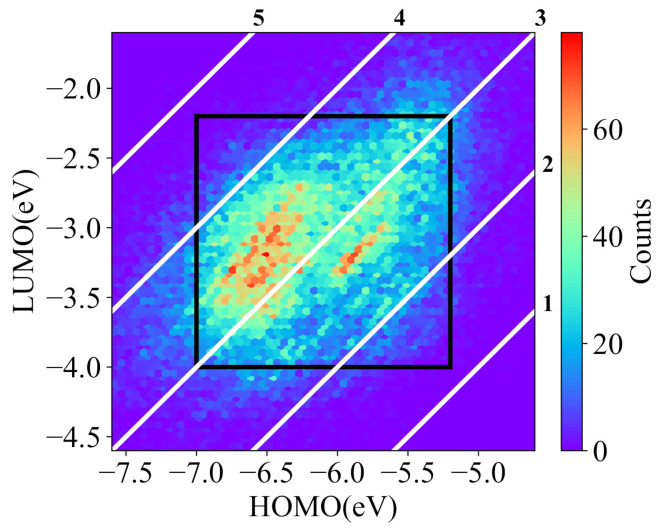
The distribution of NFAs for training the deep learning models (the white diagonals are the constant bandgap of the HOMO and LUMO).

**Figure 4 ijms-22-09099-f004:**
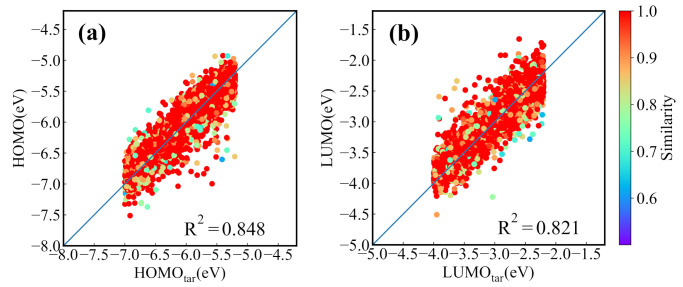
The performance of the molecular generation model for 4000 desired HOMO/LUMO energies. (**a**) The predicted HOMO energies versus targeted HOMO energies and (**b**) the predicted LUMO energies versus targeted LUMO energies of molecules generated in one generation process. The predicted values versus the targeted values of (**c**) HOMO and (**d**) LUMO energies of the best matched molecules screened out in 30 generation processes. The predicted values versus targeted values of (**e**) HOMO and (**f**) LUMO energies of the best matched new molecules screened out in 30 runs.

**Figure 5 ijms-22-09099-f005:**
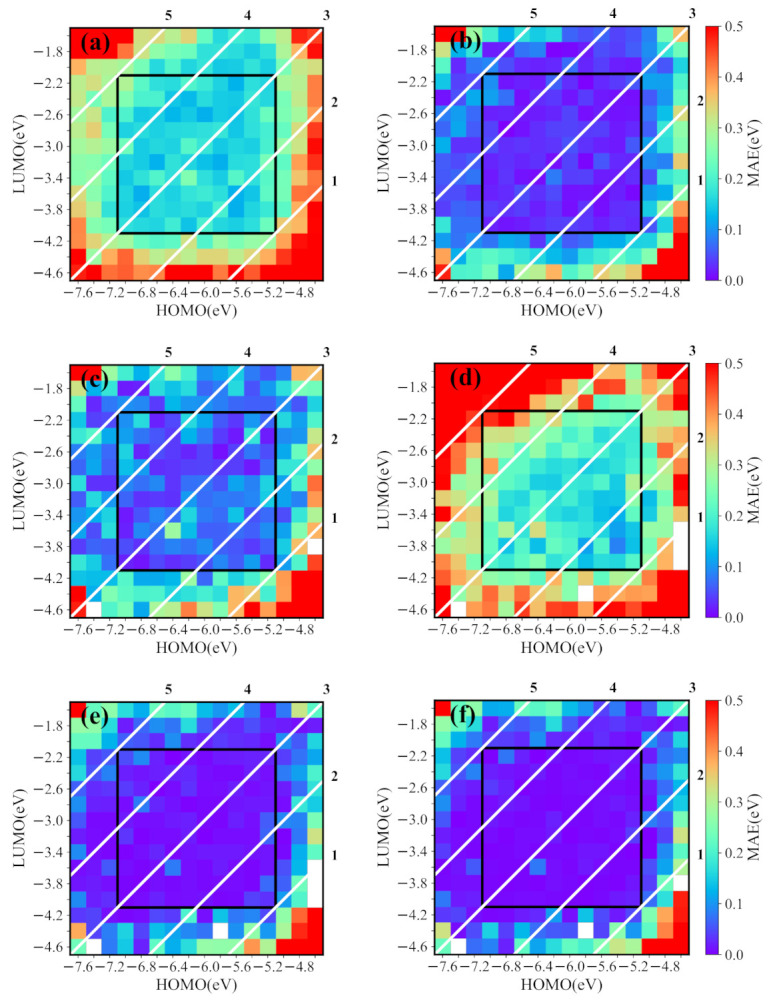
The performance of the conditional generative model for the desired HOMO/LUMO energies in a broad range. The heatmap shows the MAEs of predicted properties and targeted values for generated molecules (the white diagonals are the constant bandgap of the HOMO and LUMO). (**a**) The MAEs of molecules generated in 30 runs for desired HOMO/LUMO energies (unreasonable molecules are excluded). (**b**) The MAEs of the best matched molecules screened out in 30 runs. (**c**) The MAEs of the best matched new molecules in 30 runs. (**d**) The MAEs of all possible molecules transformed from molecules in (**c**) with the transformation rule database created by the original library. (**e**) The MAEs of the best matched new molecules in the transformed molecules. (**f**) The MAEs of the best matched new molecules transformed from molecules in (**c**) with the rule database created by the original library and our library.

**Figure 6 ijms-22-09099-f006:**
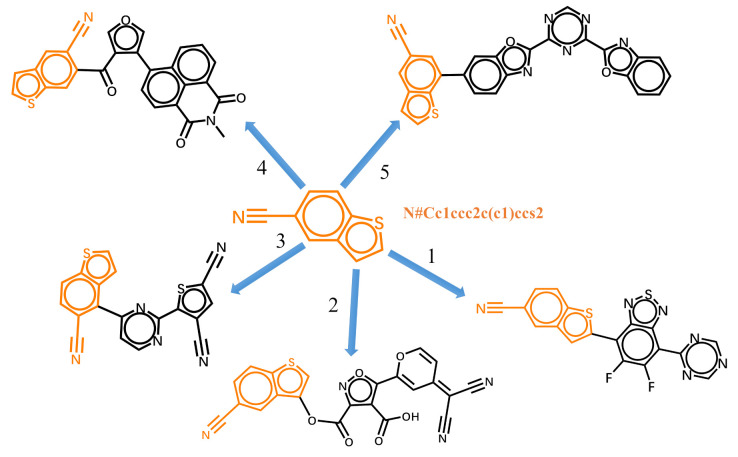
The five possible molecules generated from the given molecular fragment for the targeted HOMO energy of −6.5 eV and LUMO energy of −3.3 eV.

**Figure 7 ijms-22-09099-f007:**
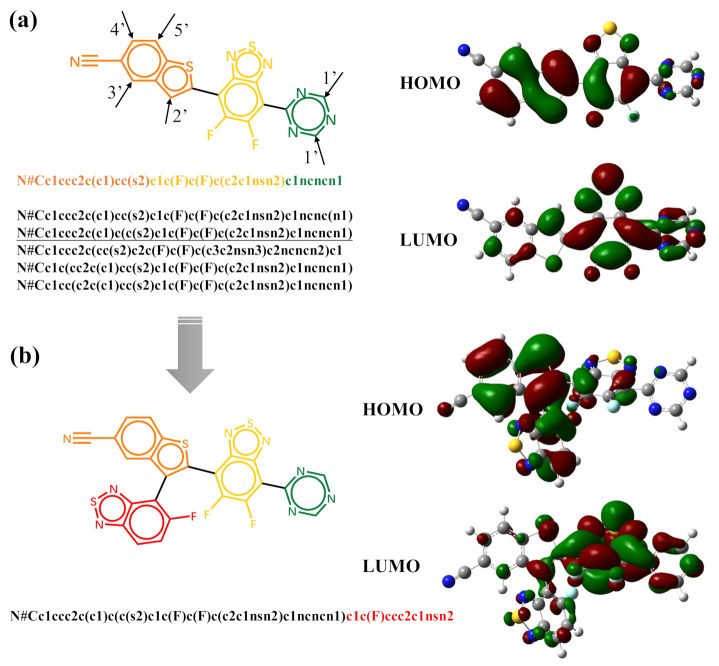
The structures of molecule (**a**,**b**) and their calculated HOMO and LUMO distributions.

**Table 1 ijms-22-09099-t001:** The SMILES strings and predicted HOMO and LUMO energies of five generated molecules.

No.	SMILES	Similarity	HOMO Energy (eV)	LUMO Energy (eV)
**1**	N#Cc1ccc2c(c1)cc(s2)c1c(F)c(F)c(c2c1nsn2)c1ncncn1	0.75	−6.614	−3.521
**2**	N#Cc1ccc2c(c1)c(cs2)OC(=O)c1noc(c1C(=O)O)C1=CC(=C(C#N)C#N)C=CO1	0.76	−6.698	−3.508
**3**	">N#Cc1ccc2c(ccs2)c1-c1nc(-c2sc(C#N)cc2C#N)ncc1	0.78	−6.541	−3.260
**4**	N#Cc1c(cc2c(c1)ccs2)C(=O)c1cocc1-c1c2c3c(ccc2)C(=O)N(C)C(=O)c3cc1	0.76	−6.468	−3.063
**5**	N#Cc1cc(c2c(c1)ccs2)c1ccc2nc(-c3ncnc(-c4oc5ccccc5n4)n3)oc2c1	0.81	−6.463	−2.881

**Table 2 ijms-22-09099-t002:** The weighted sum of each fragment from the prediction model to the molecular orbitals for molecule (a) and (b).

Mol	Orbital	Energy_calc (eV)	Energy_pred (eV)	Red	Orange	Yellow	Green
(a)	HOMO	−6.537	−6.614		0.446	0.386	0.168
	LUMO	−3.372	−3.521		0.172	0.687	0.141
(b)	HOMO	−6.499	−6.540	0.194	0.379	0.299	0.129
	LUMO	−3.240	−3.359	0.310	0.081	0.498	0.111

## Data Availability

The data presented in this study are available in Appendix A.

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
