# Peer review of "Molecular Conditional Generation and Property Analysis of Non-Fullerene Acceptors with Deep Learning"

_ijms, 2021, doi:10.3390/ijms22169099_

Round 1

Reviewer 1 Report

The authors reported a deep learning method predicting the molecular design of non-fullerene acceptors. Even though the principal idea exhibited in this manuscript is good, there are a few unclear points. Thereby, I think the manuscript can be published in Int. J. Mol. Sci. after a minor revision. Below are the comments for the authors to address. 

  1. There are a few reports related to non-fullerene acceptors with deep/machine learning. Can the authors describe what is difference compared to other articles?
  2. This work just focused on the energy levels (e.g., HOMO and LUMO), which is the core of the convolutional neural network to predict the molecular design. However, the solubility (e.g., the length of alkyl chains) and steric hindrance are also very significant for the design of non-fullerene acceptors, which were not found in the manuscript. Can the author explain why you did not consider these critical material features? This should be mentioned in the manuscript.
  3. Regarding the molecular properties and design of non-fullerene acceptors, authors need to provide more details.

Reviewer 2 Report

The manuscript "Molecular conditional generation and property analysis of non-fullerene acceptors with deep learning" by Shi-Ping Peng, Xin-Yu Yang, and Yi Zhao describes how a vast number of potential non-fullerene acceptors (NFAs) can be screened/checked using deep learning algorithms, with a specific focus on the frontier orbital, namely the HOMO and LUMO energies. Indeed, this type of research is quite important to establish the (relatively) new techniques based on machine learning in fields outside of those typically associated with machine learning, such as computer science. Furthermore, it is apparent from the vast number of tested molecules that this type of research approach possesses the potential to revolutionize research in the field of structure-property-relationships in chemistry. Therefore, the general topic of this manuscript will be quite interesting to the wide readership of the International Journal of Molecular Science. However, before publication may happen, the following points need to be addressed by the authors:

1. The authors should add more details about the topic of organic photovoltaics and non-fullene acceptors in the introduction. For instance, the potential of NFAs in OPVs should include economic aspects , and specific applications, such as semi-transparent OPVs for a use in buildings or agriculture , or the use of NFAs in organic photodetectors. Furthermore, the jump in performance observed in OPVs thanks to NFAs should be outlined a bit more in detail, mainly breaking the PCE = 15% threshold by using Y6 in 2019. Providing this context in more detail would improve the manuscript.

2. The authors should also consider to add a brief summary/introduction to the basics of machine learning, since the International Journal of Molecular Science is not mainly focused on research using machine learning, but is rather a general, chemistry focused journal. Thus, the authors could use the present article as a gateway to introduce non-experts to the field of machine learning in the scope of chemistry.

3. In Fig. 3 and 5, the authors could also add diagonals with constant bandgaps (see attached document), since more often than not, the bandgap also is an important figure when designing organic solar cells.

4. On page 10, lines 191/192, the authors mention a molecule at a specific HOMO and LUMO energy shown in Figure 5(f). It might be best to add a sign (cross or similar) to the pixel at those energies in Figure(f), as that would it make easier for the readers to locate the molecule in question.

5. Can the authors elaborate on the meaning of the green and red colored molecular fragments in Fig. 7? In the text, the orange and yellow sections were described as dominating the HOMO and LUMO energies, respectively.

6. Minor comments about expression, typos, grammatical errors, layout/style and phrasing can be found in the attached document.
